



# Investigation of adsorption/desorption behavior of small volume cylinders and its relevance for atmospheric trace gas analysis

Ece Satar[1,2], Peter Nyfeler[1,2], Bernhard Bereiter[1,2,3], Céline Pascale[4], Bernhard Niederhauser[4], and Markus Leuenberger[1,2]

[1]Climate and Environmental Physics, Physics Institute, University of Bern, Bern, Switzerland
[2]Oeschger Centre for Climate Change Research, University of Bern, Bern, Switzerland
[3]Empa, Laboratory for Air Pollution / Environmental Technology, Dübendorf, Switzerland
[4]Federal Institute of Metrology METAS, Bern, Switzerland

**Correspondence:** satar@climate.unibe.ch

**Abstract.** Atmospheric trace gas measurements of greenhouse gases are critical in their precision and accuracy. In the past 5 years, atmospheric measurement and gas metrology communities have turned their attention to possible surface effects due to pressure and temperature variations during a standard cylinder's lifetime. This study concentrates on this issue by introducing newly built small volume aluminum and steel cylinders which enable the investigation of trace gases and their

5  affinity for adsorption/desorption on various surfaces over a set of temperature and pressure ranges. The presented experiments are designed to test the filling pressure dependencies up to 30 bar, and temperature dependencies from $-10\,^{\circ}\mathrm{C}$ up to $180\,^{\circ}\mathrm{C}$ for these prototype cylinders. We present measurements of $CO_2$, $CH_4$, $CO$ and $H_2O$ using a cavity ring down spectroscopy analyzer under these conditions. Moreover, we investigated $CO_2$ amount fractions using a novel quantum cascade laser spectrometer system enabling measurements at pressures as a low as 5 mbar. This extensive dataset revealed that until pressures as low as 150

10  mbar the enhancement in the amount fraction of $CO_2$ relative to its initial value (at 1200 mbar) is limited to $0.12\,\mu\mathrm{mol\,mol^{-1}}$ for the prototype aluminum cylinder. Up to $80\,^{\circ}\mathrm{C}$, the aluminum cylinder showed superior results and less response to varying temperature compared to the steel cylinder. For $CO_2$, these changes were insignificant at $80\,^{\circ}\mathrm{C}$ for the aluminum cylinder, whereas a $0.11\,\mu\mathrm{mol\,mol^{-1}}$ enhancement for the steel cylinder was observed. High temperature experiments showed that for both cylinders irreversible temperature effects occur especially above $130\,^{\circ}\mathrm{C}$.

## 1  Introduction

Atmospheric measurements play a crucial role in understanding the global carbon cycle and its response to anthropogenic perturbation. The first atmospheric measurements of $CO_2$ were done in Hawaii in Mauna Loa in the late 1950s (Pales and Keeling, 1965). Ever since, the number of stations for atmospheric observations has increased continuously. Most of these measurements are conducted in remote areas, and with increasing number of stations it is more challenging to ensure the comparability of

20  the measurements. The coordination of the greenhouse gas measurement network is achieved by the World Meteorological Organization (WMO) through its Global Atmospheric Watch Programme (GAW). WMO also makes recommendations on the compatibility targets for the measurement stations within its network. For $CO_2$, these targets correspond to $0.1\,\mu\mathrm{mol\,mol^{-1}}$ for





the northern hemisphere, and 0.05 µmol mol$^{-1}$ for the southern hemisphere (WMO, 2018). These ambitious targets allow the interpretations of fluxes on global and continental scales, and to better distinguish the underlying processes (Rödenbeck et al., 2006; Masarie et al., 2011).

In order to ensure quality observations, the measurement systems are calibrated regularly with known standards. In addition
to careful and regular calibrations, it is important to be able to account for the instabilities which might affect the measured amount fractions of trace gases. More than a decade ago, Langenfelds et al. (2005) and Keeling et al. (2007) have reported deviations of $O_2/N_2$ and $CO_2$ in standard cylinders. The former study has suggested that diffusive fractionation is the main process, however for $CO_2$ diffusive fractionation alone was not sufficient to explain the observed enrichment. The latter study reported a downwards drift in their aluminum cylinders with respect to steel cylinders. Keeling et al. (2007) explained this
difference by conditioning wall reactions. These studies also mention leakage, regulator effects, and thermal and gravimetric fractionation as responsible processes for instabilities.

With the advances in analytical techniques and improvements in measurement uncertainties, more stringent targets of better comparability became possible. The discussion on surface effects has got attention in the last five years within both atmospheric measurement and gas metrology communities. For $CO_2$, the studies from Leuenberger et al. (2015) and Miller et al. (2015)
interpreted their findings in favor of adsorption and desorption processes. Leuenberger et al. (2015) emptied the cylinders and observed enrichments in the amount fractions of $CO_2$ due to pressure loss, whereas Miller et al. (2015) filled the cylinders and calculated mother to daughter ratios and reported losses to gas wetted surfaces. Moreover, Leuenberger et al. (2015) used Langmuir (1918) monolayer adsorption isotherm to explain the observed enrichment in $CO_2$ amont fractions. These studies were followed by another pair of studies from both communities (Brewer et al., 2018; Schibig et al., 2018). These experiments
were designed to include also cylinders with surface passivation and different water vapor content. The results of these studies confirmed that adsorption/desorption processes is at play, but also Rayleigh fractionation during high flow experiments plays a role (Schibig et al., 2018).

This study contributes to currently existing literature by presenting data on below ambient pressures and high temperatures separately. The experiments are done using two newly built small volume aluminum and steel cylinders. This paper focuses on
several issues related to gas cylinders usage: (i) pressure dependency of surface processes with respect to filling pressure, (ii) pressure relation of adsorption/desorption for very low pressure ranges, and (iii) possible effects of heating on cylinders.

## 2 Data and Methods

### 2.1 Production and filling history of the small cylinders

In order to understand adsorption/desorption effects to its full extent, we scaled down the problem. For this purpose, high
pressure (up to 130 bar) and small volume (5 L) cylinders of aluminum and steel were designed. The aluminum cylinder is made of the aluminum alloy AlMg1SiCu (EN AW-6061), and the steel cylinder is made of hardened and tempered steel (1.7218 / 25CrMo4, EN 10273). The cylinder compositions are specifically chosen such that they correspond to steel and aluminum materials commonly in use for high pressure cylinders in the atmospheric measurement community.





The 5 L cylinders are formed from raw materials without welding at the workshop of the University of Bern. Each cylinder consists of three pieces (Fig. A1a): a body part in the middle with two caps on the sides. These pieces are joined by twelve necked-down bolts on each side, and Inconel X750 seals with silver coating are placed in the caps. This setup enables to place test materials into these chambers to investigate their surface effects.

After their production at the workshop of the University of Bern, the cylinders were sent to the external firm that designed them for pressure testing and certification for usage at high pressures. Then, the cylinders underwent a cleaning procedure consisting of ultrasonic bath with a diluted solution of a mildly alkaline commercial cleaning agent (Deconex HT1201), and oven drying (Table 1a).

Each cylinder is equipped with four full-metal valves (SS-4H from Swagelok). All tubings (1/4") and connections are
made of stainless steel, while the tubings are also electropolished. At the outlet, the cylinders are equipped with a dual stage pressure regulator made of stainless steel with a polychlorotrifluoroethylene (PCTFE) seat (64-3441KA412 from Tescom). Pressure transducers are used at low (PTU-S-AC6-31AC from Swagelok), and high (PTU-S-AC160-31AC from Swagelok) pressure sides of the pressure regulators. Temperature sensors spanning a range from -35 °C to +100 °C (AF25.PT100 from Thermokon) are placed on the outer cylinder surfaces and one is placed on a pressure regulator. All measured temperature
and pressure data were read and logged by a signal converter (midi logger GL820 from Graphtec). For experiments exceeding 100 °C another digital thermometer (Greisinger GMH 3250) was used.

The filling history of the small cylinders is the following: They were filled with $N_2$ (Alphagaz 2 $N_2$) up to 2 bar (relative to ambient pressure) for background measurements (data not presented here). According to its specifications, Alphagaz 2 $N_2$ contains very low amount fractions of CO ($< 0.1$ μmol mol$^{-1}$), $CO_2$ ($< 0.1$ μmol mol$^{-1}$), $CH_4$ ($< 0.1$ μmol mol$^{-1}$) and $H_2O$
($< 0.5$ μmol mol$^{-1}$). The rest of the fillings were done using high pressure 50 l aluminum cylinders with compressed air. These aluminum cylinders (LUX3588, LUX3575, and LUX3586) are referred to as mother cylinders from here on. A scheme of the measurement setup is given in Fig. 1a. The fillings which did not exceed 6 bar were conducted directly through the pressure regulator same type as above (64-3441KA412 from Tescom), and the fillings exceeding 6 bar were done by expansion. In this case, the mother cylinder was directly connected to a small expansion volume (0.5 L) of stainless steel (316L-HDF4-500 from
Swagelok). The desired pressure in the small cylinder was achieved by repeating the expansion step several times. Bracketing each sample measurement, another aluminum cylinder of comparable material and equipment to the mother cylinder was measured to check the stability of the measurement device (LUX3579).

## 2.2 Experiments with CRDS analyzer

Both temperature and pressure experiments were conducted using these newly built cylinders. An overview of all presented
experiments and procedures are given in Table 1. The presented experiments cover in total two years, and the chronology of the experiments and procedures are essential to interpret the results.

In order to investigate pressure dependency, the aluminum cylinder was filled in five pressure levels (1.5, 6, 8, 20 and 30 bar) and the steel cylinder was filled in four pressure levels (6, 8, 20 and 30 bar) (Table 1b). Each measurement series had at least three replicates. For pressure experiments, the small cylinders were filled, and an hour was given for equilibration. Then,



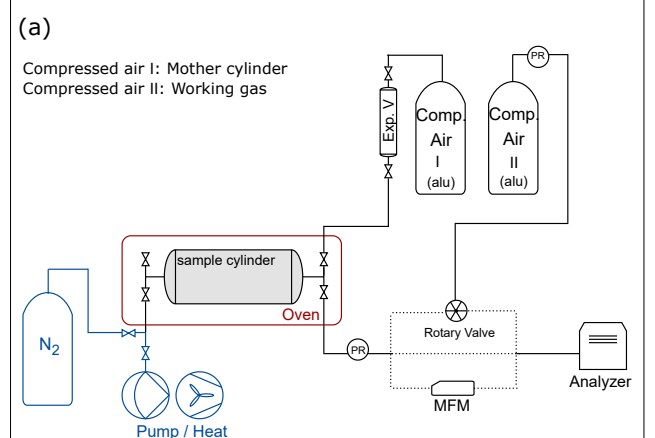
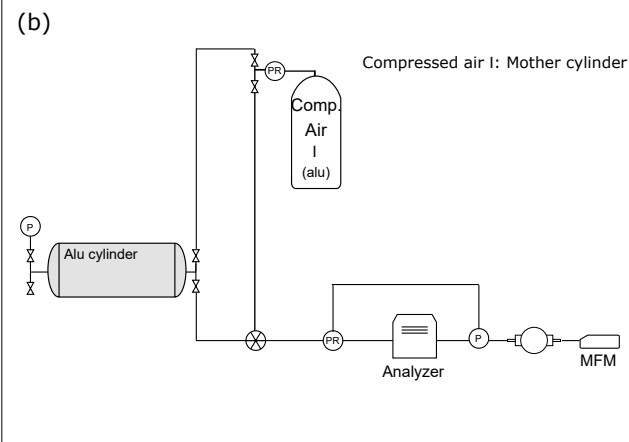

**Figure 1.** Measurement schemes: **(a)** CRDS analyzer. The sample cylinder is filled by expansion from the mother cylinder, and placed in the climate cabinet (red box) for the temperature experiments. Dotted lines show the three different pathways between the outlet of the cylinder and the inlet of the analyzer. During temperature experiments, sample gas flows through the rotary valve, during several experiments (Table 1h) sample gas flows through mass flow meter (MFM) to monitor flow. For all remaining experiments, the sample gas is directed to the analyzer with electropolished stainless steel tubing. The equipment used for the cleaning procedure applied in between experiments in Table 1e is shown in blue. **(b)** QCLAS system. The sample cylinder is filled through the pressure regulator. The inflow to the analyzer is switched between the mother and sample cylinder through the rotary valve. The loop around the analyzer shows the pressure regulation (See Sect. 2.3 for a detailed description). After the turbomolecular pump a MFM is placed to monitor outflow.

the cylinder was measured continuously with a Picarrro Cavity Ring Down Spectroscopy (CRDS) G2401 analyzer, allowing measurements of $CO_2$, $CO$, $CH_4$ and $H_2O$. In between the experiments, the cylinders were only pumped through the analyzers external pump, and the next filling was done without flushing with another gas.

In order to investigate the temperature dependency, the small cylinders were placed into a climate cabinet (ACS Challenge
5  600) at the Swiss Federal Institute of Metrology (METAS). The temperature of the cabinet was set at temperatures from –10 °C to 180 °C (Table 1c). For low temperature experiments, the temperature was set at –10 °C, 20 °C, 50 °C and 80 °C, with 30 °C increments, heated or cooled within an hour (Fig. 2a). Whereas for high temperature experiments, temperature was set at 20 °C, 80 °C, 130 °C and 180 °C, and heated or cooled within two hours (Fig. 2b). The set temperature was kept constant for four hours at each level, of which during the last 35 minutes the sample cylinder was measured. These measurements were
10  bracketed by working gas measurements (LUX3579) which did not experience any temperature changes. A multiport valve (EMT2CSD6MWE from VICI AG) was used to switch between the small cylinders and the working gas.

Moreover, we applied a further cleaning procedure during the course of the temperature experiment set (Table 1d). This corresponded to heating and pumping cycles of several hours at 180 °C for the steel cylinder; and opening, ultrasonic cleaning, and polishing for the aluminum cylinder. Firstly, the aluminum cylinder was opened and placed in an ultrasonic-bath with a





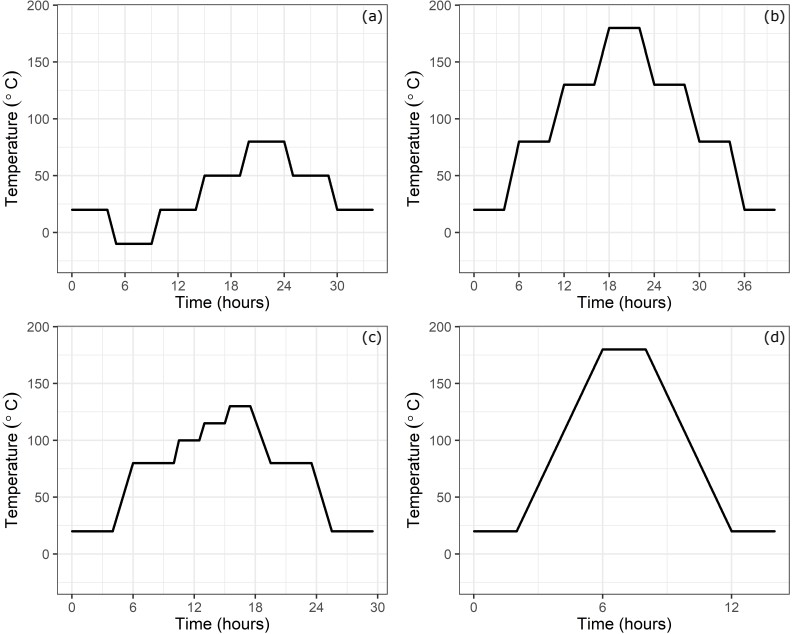

**Figure 2.** Temperature setting at the climate cabinet: **(a)** low temperature experiments, **(b)** high temperature experiments, **(c)** experiments until 130 °C and **(d)** high temperature experiments for only 180 °C.

mild detergent and tap water, however the ultrasonic bath cycles at 60 °C ended with further contamination and visible stains (Fig. A1b). To eliminate this, the two caps were polished with an organic agent, and the cylinder underwent a further cycle of ultrasonic bath with a mild detergent and tap water, followed by three cycles of cleaning with tap water and a final round with reverse osmosis water.

5    After the second cleaning procedure, to investigate the high temperature effects in more detail, we did several fillings with synthetic air and $N_2$ (Table 1e) with fewer step changes at various temperatures (Fig. 2d).

Due to the fact that heating of cylinders led to contamination, we established a cleaning procedure consisting of three pump-heat cycles of 30 minutes each in between the temperature experiments presented in Table 1e. During each cycle, cylinders were filled with 2 bar $N_2$, and while pumping, the cylinder was heated by a heat gun, where its surface temperature did not

10   exceed 60 °C. The cylinder was pumped using a dry piston vacuum pump (EcoDry M15 from Leybold) until 0.05 mbar.

For better comparison of pressure behavior after temperature experiments and cleaning, we included data from a filling pressure of 14 bar (Table 1f). These experiments were conducted within another study related to these cylinders (Satar et al., 2019). And lastly, we did several fillings of 3.5 bar (Table 1h). For these last measurements a mass flow meter (Series 358 from Analyt - MTC) was placed at the inlet or at the outlet of the analyzer in order to monitor flow.



**Table 1.** An overview of experiments and procedures included in this study in chronological order

| Cylinder | Experiment | Pressure [bar] | Number of replicates | Mother cylinder |
|---|---|---|---|---|
| **a. First cleaning** – February 2017 | | | | |
| Aluminum – Steel | Ultrasonic bath and oven dried | | | |
| **b. Pressure experiments with CRDS** – February 2017 – June 2017 | | | | |
| Aluminum | Pressure | 1.4 ... 1.5 | 4 | LUX3588 |
| Aluminum | Pressure | 5.7 ... 5.8 | 4 | LUX3588 |
| Steel | Pressure | 5.7 ... 6.0 | 4 | LUX3588 |
| Steel | Pressure | 7.3 - 7.3 - 7.5 | 3 | LUX3588 |
| Steel | Pressure | 18.6 ... 20.8 | 4 | LUX3588 |
| Steel | Pressure | 24.3 - 25.4 - 26.4 | 3 | LUX3588 |
| Aluminum | Pressure | 7.5 - 7.7 - 8.0 | 3 | LUX3588 |
| Aluminum | Pressure | 16.9 - 17.8 - 18.6 | 3 | LUX3588 |
| Aluminum | Pressure | 28.3 - 29.4 - 29.8 | 3 | LUX3588 |
| **c. Low and high temperature experiments with CRDS** – June 2017 – August 2017 | | | | |
| Aluminum | Low T: –10 °C - 80 °C | 26 | 1 | LUX3588 |
| Aluminum | High T: 20 °C - 180 °C | 23.8 | 1 | LUX3588 |
| Steel | Low T: –10 °C - 80 °C | 23.3 | 1 | LUX3588 |
| Steel | High T: 20 °C - 180 °C | 21.9 - 20.5 - 19.3 | 3 | LUX3588 |
| Steel | Up to 130 °C | 18.1 | 1 | LUX3588 |
| **d. Second cleaning** – August 2017 | | | | |
| Aluminum | Opened, ultrasonic bath, and polished | | | |
| Steel | Pump-heat cycles of several hours at 180 °C | | | |
| **e. High temperature experiments with CRDS after cleaning** – September 2017 | | | | |
| Aluminum* | After cleaning: 20 °C - 180 °C | 4.81 | 1 | LUX3588 |
| Steel | After cleaning: 20 °C - 180 °C | 4.4- 4.5 - 4.6 | 3 | LUX3588 |
| Aluminum* | High T: 20 °C - 180 °C ** | 4.5 - 4.6 | 2 | Synthetic air |
| Steel* | High T: 20 °C - 180 °C | 4.6 | 1 | Synthetic air |
| Aluminum* | High T: 20 °C - 80 °C - 180 °C | 5.0 | 1 | $N_2$ |
| Steel* | High T: 20 °C - 80 °C - 180 °C | 5.1 | 1 | $N_2$ |
| **f. Pressure experiments with CRDS** – September 2018 | | | | |
| Aluminum | Pressure | 13.9 – 13.9 | 2 | LUX3575 |
| **g. Experiments with QCLAS** – October 2018 | | | | |
| Aluminum | Pressure | 0.2 | 6 | LUX3575 |
| **h. Pressure experiments with CRDS** – November 2018 – January 2019 | | | | |
| Aluminum | Pressure | 3.3 ... 3.7 | 10 | LUX3575 |
| Aluminum with MFM*** | Pressure | 3.4- 3.8- 4.2 | 3 | LUX3575 |

*Fill-pump-heat cycles with $N_2$ prior to filling
**One run with High T: 20 °C - 80 °C - 180 °C
***MFM: Mass flow meter





In order to compare different datasets, measured amount fractions were subtracted from the mean of the first hour of measurements for each run. Then, 5-minute means of these differences were calculated in order to eliminate instrumental noise. All reported values in this study from the CRDS analyzer are differences of amount-of-substance fractions and denoted by $\Delta CO_2$, $\Delta CO$, $\Delta CH_4$ and $\Delta H_2O$.

## 2.3 Experiments with dual Quantum Cascade Lasers Absorption Spectrometer (QCLAS)

In order to understand the surface processes to its full extent, the aluminum cylinder was tested at pressures as low as 5 mbar using a novel analyzer (Table 1g). These measurements were conducted at the Swiss Federal Laboratories for Material Science and Techology (Empa). The analytical approach is based on direct absorption spectroscopy using Quantum Cascade Lasers (QCLs) (Nelson et al., 2008; McManus et al., 2011). The spectrometer used in this study is built within the deepSLice project for measuring air samples of very small volumes such as 1-2 mL STP extracted from ice-cores, and is under further development. The analyzer enables simultaneous measurements of $CO_2$, $CH_4$ and $N_2O$ amount fractions and the isotopic signature of $CO_2$. In order to cover all four target species in mid-infrared spectral range, the system is equipped with two quantum cascade lasers emitting at 4.3 and 7.7 μm, respectively. Furthermore, the analyzer is optimized for low gas pressures (∼5 mbar) in its cell to cope with the small amount of sample available from ice samples.

Figure 1b shows the measurement setup for these experiments. We aimed to cover the range from atmospheric to subatmospheric pressures until the cell pressure of 5 mbar. Therefore, we filled the aluminum cylinder to 200 mbar (relative to ambient pressure) using the same type of pressure regulator as in Sect. 2.1. The cylinder was evacuated using a turbo pump at the end of the line. In order to eliminate instrumental drift, the mother cylinder was measured for five minutes every 15 minutes bracketing the measurements from the sample cylinder. The multiport valve (EUTA-SD6MWE from VICI) allowed switching between the mother cylinder and the small aluminum cylinder. After the rotary valve, the gas entering the measurement cell was controlled by a pressure controlling loop using a proportional valve (EV-P-10-0925 from Clippard) coupled with a controller (JumodTron316), and a pressure transmitter (PAA-35X Series from Keller) after the cell and before the pump. This control loop enabled measurements of the sample cylinder without a pressure regulator after the outlet of the cylinder, and provided constant flow conditions at a set cell pressure of 5 mbar. The mother cylinder underwent the same pressure control loop, however for the mother cylinder a pressure regulator was used to preset the supplied pressure close to atmospheric pressure. After the turbo pump, a mass flow meter (Series 358 from Analyt - MTC) was placed in order to monitor the flow out of the analyzer.

## 3 Results

### 3.1 Data analysis

### 3.1.1 Pressure experiments with CRDS

Since adsorption/desorption processes are pressure dependent, we aimed to cover the widest pressure range using a CRDS analyzer. However, this is not trivial since our measurements started from pressures as high as 30 bar, which was reduced





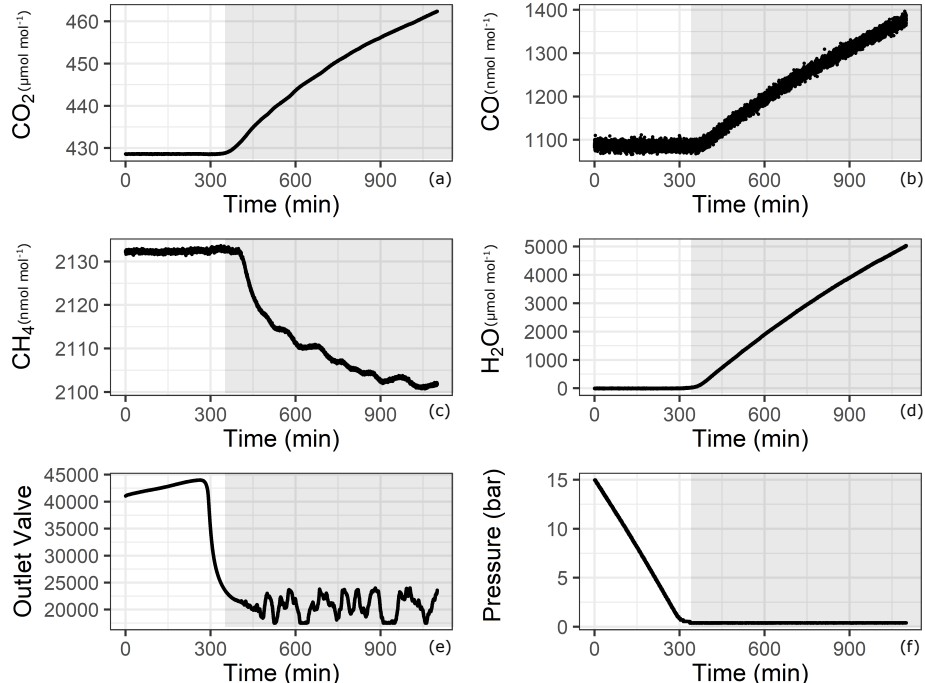

**Figure 3.** Analyzer response with respect to time for **(a)** $CO_2$, **(b)** CO, **(c)** $CH_4$, **(d)** $H_2O$, **(e)** Outlet Valve and **(f)** Pressure in the cylinder. Shaded areas correspond to depletion of sample flow.

by a pressure regulator to a pressure of 30 mbar (relative to ambient pressure) before the analyzer inlet. During the course of evacuation, the high pressure side of the regulator approached atmospheric pressure, and towards the end of the experiment, the regulator had no pressure difference to regulate. Under conditions of very low to no-flow, the CRDS analyzer started to show an effect of increasing amount fractions of CO and $CO_2$, and $H_2O$. This effect is illustrated in Fig. 3 for one of the experiments

5   with 14 bar in the aluminum cylinder (Table 1f). The increase is proven not to be due to a leak in the measurement line or in the analyzer, since the measured amount fraction of CO (Fig. 3b) showed an increasing trend even when the initial amount fraction in the cylinder was 10 times higher than for laboratory air. Some possible reasons for this effect might be thermal diffusion or outgassing of parts in CRDS analyzer close to the cavity. Indeed, there exists a temperature gradient between the cavity (45 °C) and the cylinder (room temperature around 25 °C). However, it is unlikely that thermal diffusion is responsible for this

10  enhancement, since both molecules with lower molecular weights such as $H_2O$ and CO, and with higher molecular weights such as $CO_2$ accumulated at the high temperature side. Moreover, the analyzer showed the same behavior when the cylinder was disconnected. Outgassing of materials close to the cavity seems to be the most likely reason for this increase, and thus it was hard to locate the exact responsible component. Even though this increase remains unclear, it is obvious that this effect is neither related to the cylinder, nor to the pressure regulator, but is originating from the analyzer. Therefore, a cut-off point for

15  the datasets is needed according to valid criteria. A possible criterion to set a cut-off point is using the CO measurements, and


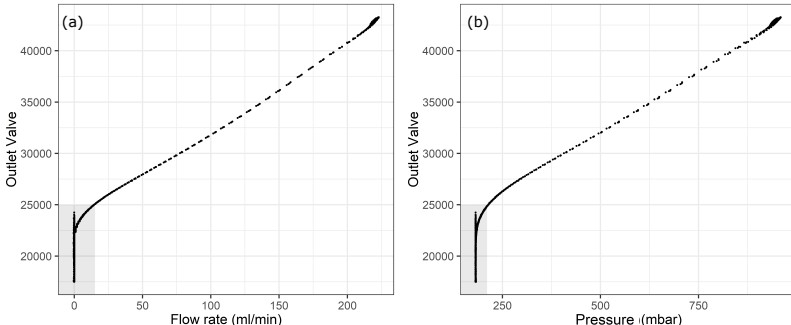

**Figure 4.** Flow parameters into the analyzer with respect to the internal parameter outlet valve **(a)** Flow rate, and **(b)** Pressure at the outlet of the pressure regulator. Shaded areas correspond to depletion of sample flow.

set the end point when CO amount fraction starts to increase. Another option would be setting a minimum inflow to the analyzer limiting the residence time in the cavity. Lastly, applying a method using the correlation between an internal variable of the analyzer (outlet valve) and measured variables (low pressure reading of the pressure regulator) would lead to a reasonable cut-off point. All methods have their advantages and disadvantages, i.e., criteria based on CO measurements or pressure reading

suffer from the quality of the measurement and its lower precision. In this section we focus on the flow criteria. Detailed information on the remaining two methods and their application are provided in the supplementary material.

The presented method aims to provide an end point which ensures sufficient flow through the analyzer. It is important to note that according to the datasheet of the G2401 CRDS analyzer, the inlet pressure is suited to be as low as 400 mbar. A possible approach would be using the pressure at the inlet of the analyzer from the reading of the pressure transducer at the low

pressure side of the pressure regulator. Since these pressure values are measured relative to atmospheric pressure, we preferred the alternative of linking our measurements to the output of an internal variable of the analyzer called the "outlet valve". This proportional valve controls the gas amount, respectively the pressure in the cavity. For the Picarro CRDS G2401 used in these experiments, the maximum value of the outlet valve is 65000 corresponding to fully open, and the minimum value is 17500 corresponding to fully closed. In case of high gas flow, the outlet valve opens and more gas is pumped out of the cavity, whereas

when the inflow to the instrument decreases, the outlet valve closes in order to keep the gas in the cavity and avoid pressure decrease. This regulation is achieved in such a way that the cell pressure is controlled to $140 \pm 0.15$ Torr ($186.65 \pm 0.20$ mbar).

The analyzer response does not correspond to reasonable values when the outlet valve values are lower than around 21500. This lower limit of the outlet valve can be validated for each measurement run individually by the first method we applied using the CO amount fraction increase (Supplementary material). Since our measurements were already in a region out of the

specification range of the analyzer, we aimed to validate the cut-off point in a more conservative manner than relying on the CO measurements. As an alternative, we chose to monitor the flow rate into / out of the analyzer. In order to achieve this, we placed a mass flow meter (MFM) either just before the analyzer inlet or after the pump at the outlet of the analyzer (Table 1h). In Fig. 4, the outlet valve value is shown against the flow and the pressure at the inlet of the instrument. Here, shaded areas





show a possible cut-off point for the experiments at 25000 for the outlet valve value with a non-zero inflow to the instrument. Since our aim was to find the minimum inflow to the instrument, and corresponding residence time, we estimated the amount of sample air in the cavity of the analyzer as 5.6 mL STP, by inserting 140 Torr, 45 °C, and 35 mL for cavity pressure, temperature and volume into the ideal gas equation. At STP conditions and 25000 outlet valve value, flow rate into the analyzer was 15 mL

5    min$^{-1}$, which accounts to flushing of the cavity less than every 30 seconds. Since we reported our results in 5-minute means, we allowed at least 10 times the residence time of the cavity to flush probable outgassing effects. In terms of pressure this cut-off point corresponded to 214 mbar, which was significantly lower than 400 mbar. In order to validate the selected cut-off point, we used an independent measurement system (Sect. 2.3), the results at low pressures are presented in the following section.

10   ### 3.1.2   Pressure experiments at low pressures using QCLAS

We have done six fillings for the empty aluminum cylinder, and three fillings with steel loading inside the aluminum cylinder (data not presented in this study). From these six fillings, only one has resulted in reasonable data. Other experiment runs have suffered from various problems such as data acquisition failure, and setting the pressure regulator of the mother cylinder to sub-atmospheric pressures.

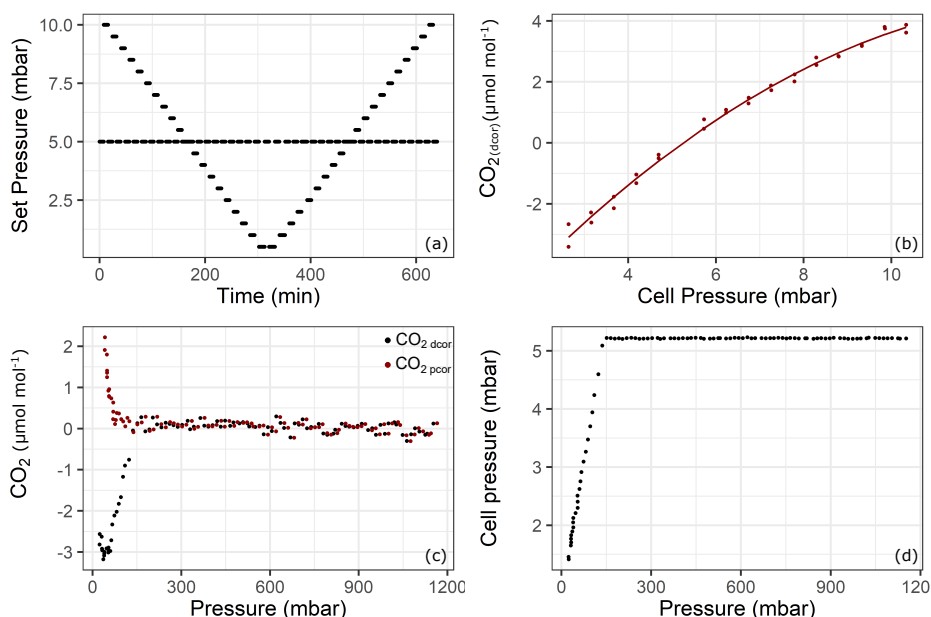

**Figure 5.** Results of the pressure experiment using QCLAS. The top panel shows the results from the cell pressure experiment: **(a)** Set pressure, **(b)** $CO_2$ response to changing cell pressure. The lower panel shows the experiment using the sample aluminum cylinder: **(c)** drift corrected $CO_2$ amount fractions (black) and pressure corrrected $CO_2$ amount fractions (red), and **(d)** cell pressure changes during the emptying of cylinder from 1200 mbar to 15 mbar.





Even though the system accounts for pressure broadening effects in the spectral analysis, there remains a pressure dependence if the cell pressure during a sample measurement is not the same as during a standard (mother cylinder) measurement. Since the pressure in the sample cylinder decreased over the experiment but the mother cylinder provided constant pressure, the cell pressure of the samples started to fall below the targeted 5 mbar cell pressure (at about 150 mbar absolute pressure in

the sample cylinder - Fig. 5d). In order to correct for this effect, the following experiment was performed: Starting from 10 mbar until 0.5 mbar with 0.5 mbar steps and back again, the cell target pressure was changed every seven minutes, while in between each of these steps, the pressure was set to the standard target pressure of 5 mbar (Fig. 5a). The measurements were first corrected for drift by subtracting the standard measurement at 5 mbar from the "sample" measurements at each pressure. Then, the following second-order polynomial was fitted to the data (Fig. 5b) to derive a cell pressure correction function:

$$x_{dcor} = a \cdot p_{set}{}^2 + b \cdot p_{set} + c \tag{1}$$

where, $x_{dcor}$ is the drift corrected amount fraction of substance, $p_{set}$ is the set pressure for the measurement cell, and $a$, $b$ and $c$ are the coefficients of the fit. The coefficient of determination ($r^2$) of the fit is 0.99. Finally, drift corrected measurement data from the fillings were corrected for pressure according to:

$$x_{pcor} = x_{dcor} - (p - p_{std}) \cdot b - (p^2 - p_{std}{}^2) \cdot a \tag{2}$$

where, $p$ is the cell pressure during the measurement and $p_{std}$ is the cell pressure during the neighboring standard measurement. In Fig. 5c, the drift corrected and pressure and drift corrected amount fraction difference relative to the mother cylinder with respect to cylinder pressure is shown. At the point where the cell pressure started to fall below the target pressure (150 mbar - Fig. 5c), the corrected data showed continuous progression, suggesting that the pressure correction is valid at that point. Towards the end of the experiment a relatively sharp increase up to 2.2 µmol mol$^{-1}$ was observed. Note, however, that this

increase occurred at cell pressures below 2.5 mbar, where the pressure correction function needed to be extrapolated. Also note that during the course of the experiment the volumetric flow through the analyzer amounted to no less than 3 mL min$^{-1}$.

However, the standard deviation of the measurements were 0.12 µmol mol$^{-1}$ over the first 24 hours before any effect has taken over (Fig. 5c). This relatively high noise was due to the lack of sensitivity of the pressure controlling loop employed (Sec. 2.3). For the amount fraction calculations, the software uses the cell pressure reading which is more sensitive, resulting in

more noisy measurements than the analyzer would achieve under its regular operation mode. Optimal parameter settings could not be found using autotuning functionality of the controller.

## 3.2  Experiments for filling pressure dependency using CRDS

In order to test for filling pressure dependency, fillings were done at various pressure levels. In Fig. 6 amount fraction differences with respect to the beginning of the experiment are plotted against the pressure in the sample cylinders. For each species,

three subplots are presented, these are grouped according to cylinder material and cylinder property, namely steel before heating (Table 1b), and aluminum before (Table 1b) and after heating (Table 1f-h). Each subplot contains a set of experiments corresponding to various pressure levels. At each pressure step, there exists at least three replicates with the exception of alu-



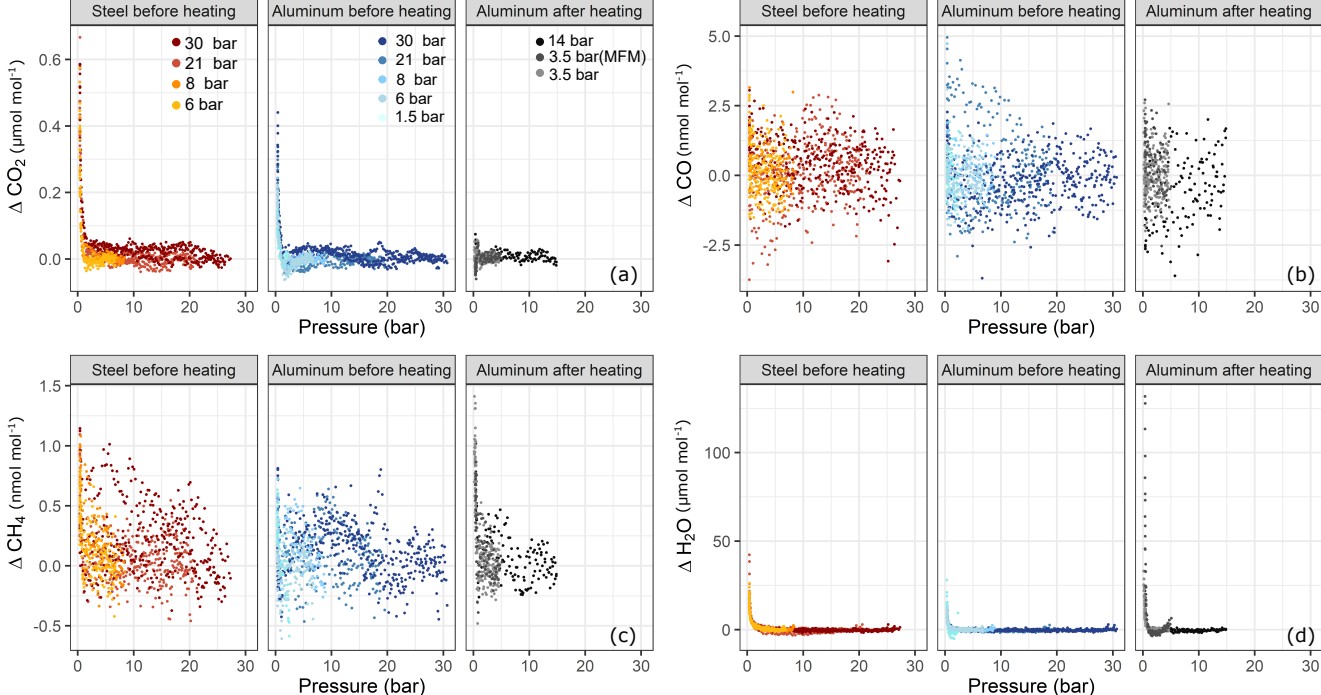

**Figure 6.** Amount fraction differences relative to the starting amount fractions during the course of each experiment with respect to pressure for **(a)** $\Delta CO_2$, **(b)** $\Delta CO$, **(c)** $\Delta CH_4$ and **(d)** $\Delta H_2O$. The subplots are grouped according to cylinder material and cylinder property: steel and aluminum before heating, and aluminum after heating (Table 1b-f-h). Each subplot contains a set of pressure steps denoted by different colors. Each of the colored series comprises replicates at that corresponding pressure step.

minum 14 bar after heating having only two replicates. For $CO_2$ (Fig. 6a) and $H_2O$ (Fig. 6d), a clear effect with decreasing cylinder pressure was observed, whereas such dependency for CO and $CH_4$ was very limited and not significant.

Figure 7 shows an overview of all pressure levels. Similarly to Fig. 6, we calculated the amount fraction differences from the initial amount fractions, and selected the maximal difference. This maximal difference was found in all cases towards the

5    end of the measurements. We interpreted this enhancement as desorption of the molecules previously adsorbed to the cylinder surface. For $CO_2$, the steel cylinder clearly showed higher enhancements at the end of the experiments corresponding to a mean of $0.49 \pm 0.03\ \mu mol\ mol^{-1}$ over the filling pressure range from 6 to 30 bar. Moreover, the aluminum cylinder showed a clear linear increase with respect to filling pressure in the final amount fractions. These corresponded to $0.14 \pm 0.02\ \mu mol\ mol^{-1}$ and $0.38 \pm 0.03\ \mu mol\ mol^{-1}$ for the lowest (1.5 bar) and the highest (30 bar) pressure step, respectively. This is a clear indication of

10   physical adsorption: The higher the filling pressure, the more $CO_2$ molecules adsorb to the cylinder surface, and with decreasing pressure in the cylinder, $CO_2$ molecules leave the surface, and mix back into the gas-phase, which results in an enhancement of the measured amount fractions. The aluminum cylinder was probably not yet in a pressure range where most of its available sites for adsorption were saturated. A linear relationship was observed with a slope of $0.01\ \mu mol\ mol^{-1}\ bar^{-1}$. However, after



**Figure 7.** Filling pressure dependency of adsorption process for aluminum and steel cylinders for **(a)** $\Delta CO_2$, **(b)** $\Delta CO$, **(c)** $\Delta CH_4$ and **(d)** $\Delta H_2O$. y-axis shows the maximal amount fraction difference relative to the initial amount fraction. Each boxplot shows the mean of the maximal amount fractions of the replicates in the center of the box denoted by a square and the median is denoted by a horizontal line.

the second cleaning procedure (Table 1d), this behavior was no longer apparent. After heating (Sect. 3.3), the pressure effect decreased to less than 0.1 µmol mol$^{-1}$ for $CO_2$, whereas 14 bar during the initial conditions would have corresponded to 0.25 µmol mol$^{-1}$ following the linear relationship. For measurements after heating, the aluminum cylinder showed a downwards trend in the amount fractions of $CO_2$ when compared to all other measurements (Supplementary material). This trend is most

5 likely related to the time lag of the outlet valve response to decreasing pressure in the cylinder. A correction was applied to account for this effect, however the measurements still show a slight decrease of 0.05 µmol mol$^{-1}$ which is larger than the standard deviation of the $CO_2$ measurements of 0.02 µmol mol$^{-1}$. More information on this correction is presented in the Supplementary material. For all other measurements such correction was not necessary since the desorption of $CO_2$ overcomes the instrumental artefact.





For both cylinders over the tested pressure range, the observed surface effects were less than 5 nmol mol$^{-1}$ and 1.5 nmol mol$^{-1}$, for the species CO (Fig. 6b) and CH$_4$ (Fig. 6c), respectively.

## 3.3 Temperature experiments with CRDS

### 3.3.1 Low temperature experiments (–10 °C to 80 °C)

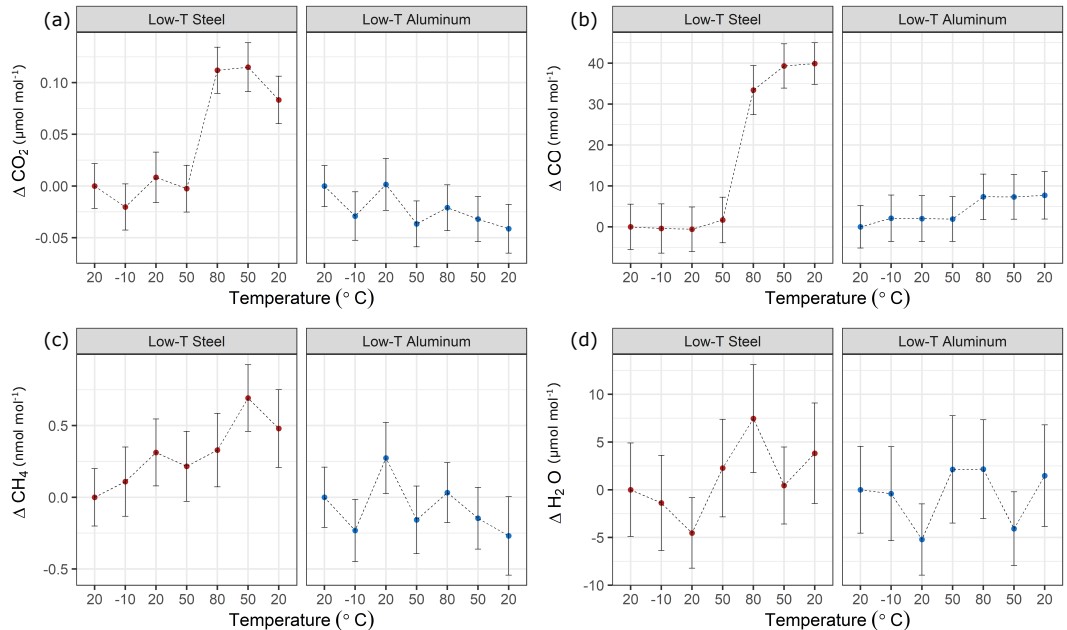

**Figure 8.** Temperature experiments until 80 °C using steel (red) and aluminum (blue) cylinder for species **(a)** CO$_2$, **(b)** CO, **(c)** CH$_4$ and **(d)** H$_2$O. x-axis corresponds to a temperature cycle, whereas the y axis shows the amount fraction differences relative to the measurements at 20 °C. Error bars indicate standard deviation of the measurements included in the mean.

5    In a first step, temperature effects at lower temperatures were investigated (Table 1c). In these experiments, the temperature of the climate cabinet was set from –10 °C to 80 °C (Sect. 2.2 and Fig. 2a). Figure 8 shows these low temperature experiments up to 80 °C, where each x-axis corresponds to a temperature cycle, and the y-axis shows the amount fraction differences relative to the begining of the experiments at 20 °C. During the low temperature experiments, the changes in amount fraction were minimal. For the aluminum cylinder, temperature effects were not significant for CO$_2$, CH$_4$ and H$_2$O, whereas for CO a slight

10    step change in amount fraction was observed. The difference between 80 °C and the first measurements at 20 °C corresponded to 7 nmol mol$^{-1}$. This difference was only marginally significant compared to the standard deviation of the CO measurements of 5 nmol mol$^{-1}$. However, it is important to note that the increase in the amount fraction was consistent even when the cylinder was cooled down back to 20 °C. In contrast to the aluminum cylinder, the steel cylinder showed higher effects for CO$_2$ and CO. When the cylinder was heated from 50 °C to 80 °C, the enhancement in the amount fraction of CO$_2$ and CO were 0.11





μmol mol$^{-1}$ and 33 nmol mol$^{-1}$, respectively. For both species, the observed increase was five to six times higher than the standard deviation of the measurements. $CO_2$ showed a slight response to cooling, whereas CO amount fraction remained stable and unresponsive to cooling back to 20 °C.

### 3.3.2 High temperature experiments (20 °C to 180 °C)

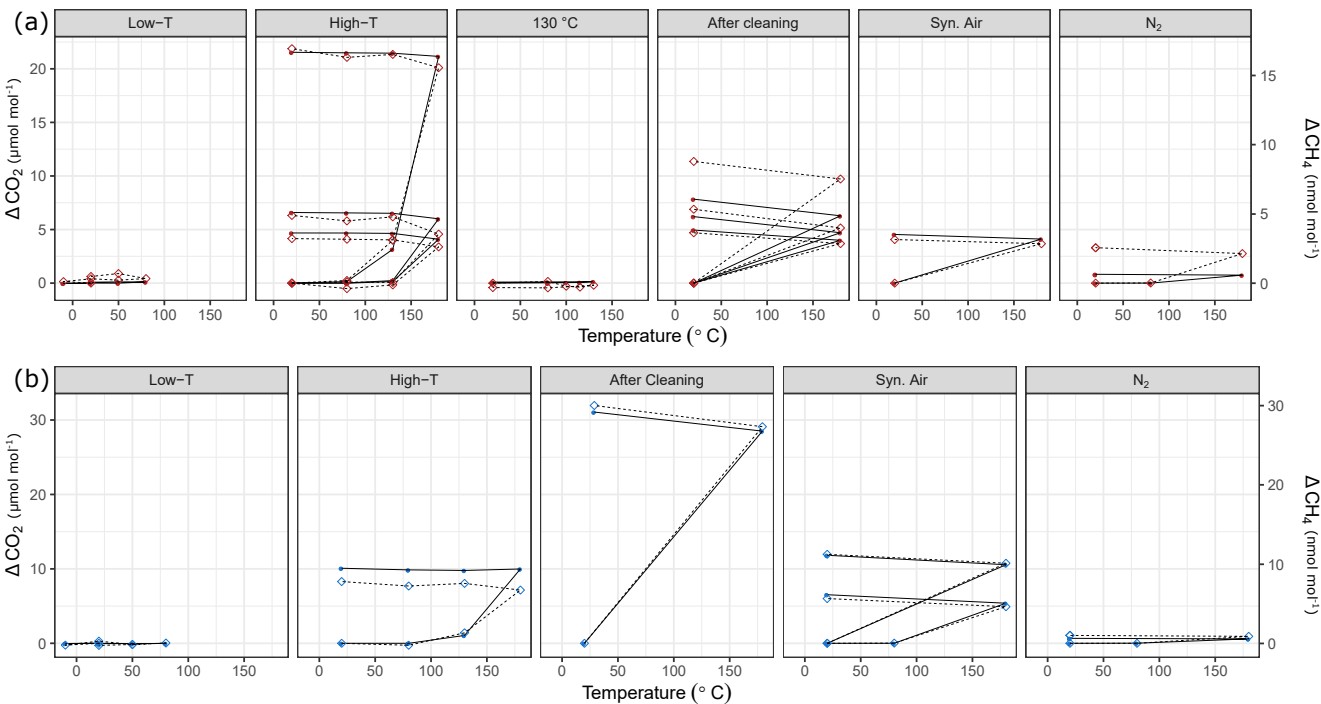

**Figure 9.** Temperature experiments using **(a)** steel (red), and **(b)** aluminum (blue) cylinder in chronological order for species $CO_2$ and $CH_4$. Filled circles correspond to left y-axis ($CO_2$), whereas open diamonds correspond to right y-axis ($CH_4$). See Table 1 for details of the temperature cycles. In each subplot temperature cycles are connected with solid lines for $CO_2$, and with dashed lines for $CH_4$. y-axes quantify the differences from the amount fractions measured at 20 °C ($\Delta CO_2$ and $\Delta CH_4$).

5      In a further step, our aim was to understand the effects of temperature variations on cylinders to its full extent due to the discrepancy we have observed previously in big cylinders (not presented here). In order to investigate temperature effects in its extremes, we heated the cylinders up to 180 °C (Table 1c). In Fig. 9 and Fig. 10 all temperature experiments are shown. Please note that at each y-axis, the amount fractions differences relative to initial conditions at 20 °C are shown ($\Delta CO_2$, $\Delta CO$, $\Delta CH_4$ and $\Delta H_2O$). Each x-axis corresponds to temperature, and each temperature cycle is in counter clockwise direction connected

10    by solid or dashed lines. In order to highlight the well-correlated response of the measured species, we scaled the second y-axis using the factor derived from the correlation between the species using reduced major axis regression separately for aluminum and steel cylinders. Slopes and the coefficients of determination ($r^2$) are given in Table 2.





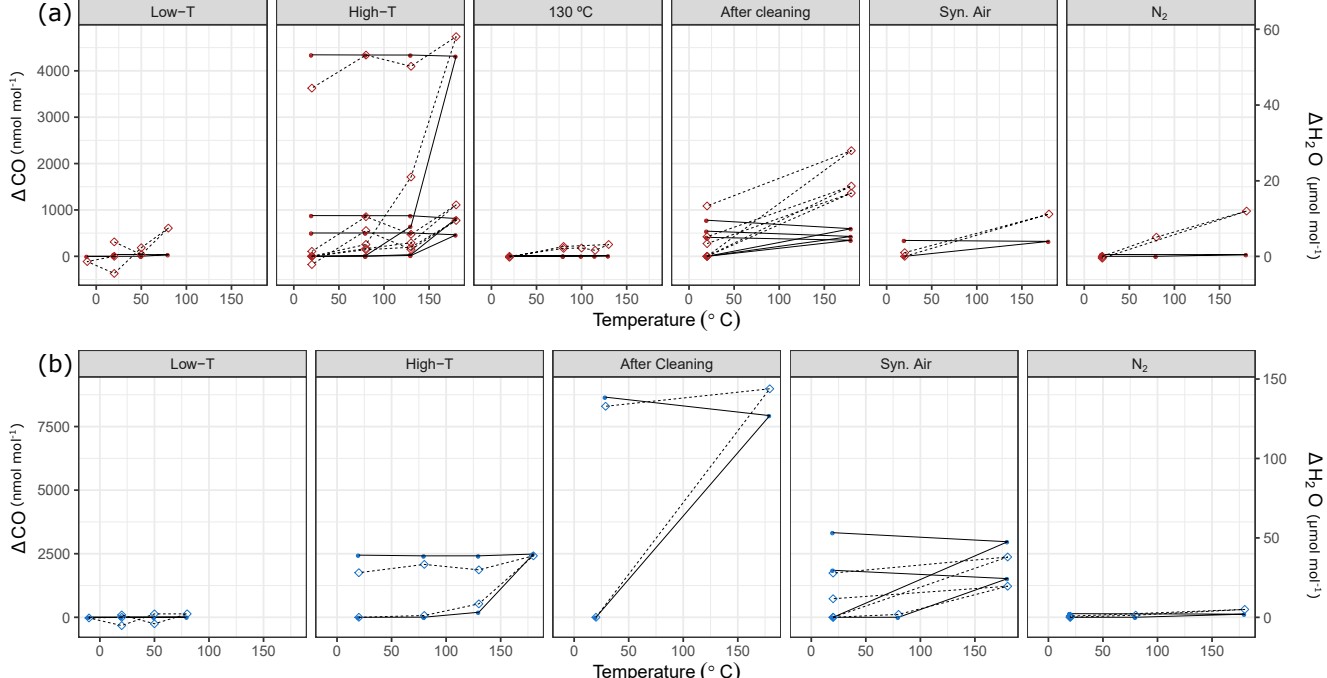

**Figure 10.** Temperature experiments using **(a)** steel (red), and **(b)** aluminum (blue) cylinder in chronological order for species CO and $H_2O$. Filled circles correspond to left y-axis (CO), whereas open diamonds correspond to right y-axis ($H_2O$). See Table 1 for details of the temperature cycles. In each subplot, temperature cycles are connected with solid lines for CO, and with dashed lines for $H_2O$. y-axes quantify the differences from the amount fractions measured at 20 °C ($\Delta CO$ and $\Delta H_2O$).

**Table 2.** Reduced major axis regression slopes and the coefficients of determination among species for temperature experiments

| Cylinder | $CO_2$ - $CH_4$ | | CO - $H_2O$ | |
|---|---|---|---|---|
| | slope | $r^2$ | slope | $r^2$ |
| Aluminum | 1064.81 | 0.99 | 624.28 | 0.96 |
| Steel | 1292.28 | 0.97 | 815.77 | 0.83 |

During the first high temperature experiments, the steel cylinder showed enhancements as high as 21.54 µmol mol$^{-1}$, 16.9 nmol mol$^{-1}$ and 4345.5 nmol mol$^{-1}$ for $CO_2$, $CH_4$ and CO respectively. However, in the second high temperature cycle these values reduced significantly, and further decreased in the third temperature cycle amounting to 4.68 µmol mol$^{-1}$, 3.2 nmol mol$^{-1}$ and 403.2 nmol mol$^{-1}$, for $CO_2$, $CH_4$ and CO respectively. With the exception of the first temperature cycle, the

5    enhancements took place after 130 °C and the changes in $H_2O$ amount fraction were reversible with temperature.





In order to highlight that these effects occurred only at higher temperatures, a temperature experiment was done up to 130 °C only for the steel cylinder (Fig. 2c). The results supported that the mechanism behind these enhancements is activated rather at temperatures higher than 130 °C. Compared to high temperature experiments (180 °C), the amount fractions increase after heating until 130 °C were an order of magnitude lower being 0.1 µmol mol$^{-1}$ and 14.0 nmol mol$^{-1}$, for $CO_2$ and CO

respectively. During the 130 °C cycle, the amount fractions of $CH_4$ remained unchanged, whereas $H_2O$ showed a reversible temperature response with a slight enhancement of 3 µmol mol$^{-1}$ at the highest temperature. After cooling back to 20 °C, this difference was not observed anymore.

During high temperature experiments, the aluminum cylinder behaved similarly to the steel cylinder with less drastic enhancements in the amount fractions of $CO_2$, $CH_4$ and CO corresponding to 10.82 µmol mol$^{-1}$, 7.8 nmol mol$^{-1}$ and 2444.1

nmol mol$^{-1}$. We suspected that these enhancements were related to reactions with the cleaning agent used in the ultrasonic bath (Table 1a), and it was still present on the cylinder surface. As explained in Sect. 2.2, after the first high temperature experiments, it was decided to apply a further cleaning procedure (Table 1d) for the cylinders to eliminate the traces of the cleaning agent.

### 3.3.3   High temperature experiments after the second cleaning procedure (20 °C to 180 °C)

After the second cleaning procedure, the programmed temperature ramps were optimized to focus only on 20 °C and 180 °C (Fig. 2d and Table 1e), and the reversibility of the enhancements. For the steel cylinder, the applied procedure of flushing and heating did not result in a significant improvement in the amount fractions measured at 180 °C, all species were observed similarly to high temperature experiments (Sect. 3.3.2). The discrepancy between the last high temperature and first run after cleaning might be due to the steepness of the set temperature curve which might affect the mechanism of this enhancement

process. However, the basic response remained unchanged, in which $CO_2$, $CH_4$ and CO showed an increase in amount fraction with decreasing intensity at each consecutive cycle, and that $H_2O$ showed a reversible behavior.

The cleaning procedure applied on the aluminum cylinder (Table 1d) made no improvement on preventing the enhancements of the species, on the contrary, it has worsened the previous situation, amounting to differences three to four times higher: 31.07 µmol mol$^{-1}$, 30.0 nmol mol$^{-1}$, 8655.4 nmol mol$^{-1}$ and 132.91 µmol mol$^{-1}$ for $CO_2$, $CH_4$, CO and $H_2O$, respectively.

In order to reveal more hints of the underlying mechanism of the observed differences, several runs were done with synthetic air and $N_2$ (Table 1e). The fillings with synthetic air were intended to ensure that the natural compressed air used for the measurements did not include gases such as volatile organic compounds which can undergo reactions at high temperatures. The fact that both synthetic air and the compressed air filling show similar differences in amount fractions after heating to 180 °C indicates that the responsible compounds did not come from natural compressed air. In addition to the measurements

with the Picarro CRDS analyzer, we sampled gas from cylinders before and after heating in the oven, and analyzed it by a Gas Chromatography with Flame-Ionization Detector technique (GC-FID) (TurboMatrix350 - Clarus500 from PerkinElmer) at the METAS Gas Laboratory. The adsorbent used was suitable for airborne C4-C8 compounds. The chromatograms showed no distinct peaks, and showed no difference between the samples before and after temperature experiments.





At a second step, the cylinders were filled with $N_2$ (Table 1e). A minimal difference in the amount fractions compared to all other high temperature runs were measured after the fillings with $N_2$. For the steel cylinder, the species $CO_2$, $CH_4$ and CO showed enhancements of 0.82 µmol mol$^{-1}$, 2.56 nmol mol$^{-1}$ and 39.7 nmol mol$^{-1}$, respectively. The amount fraction of $H_2O$ was measured as 11.96 µmol mol$^{-1}$ at 180 °C with reversible response to heating/cooling, and thus showed a non-significant

difference when cooled back to 20 °C. For the aluminum cylinder the observed differences for $N_2$ filling was 0.63 µmol mol$^{-1}$, 1.0 nmol mol$^{-1}$ and 143.9 nmol mol$^{-1}$, for $CO_2$, $CH_4$ and CO, respectively. Similar to the steel cylinder, $H_2O$ measurements were reversible, and amounted to a maximum of 4.96 µmol mol$^{-1}$. However, these results should be interpreted carefully since small changes in amount fractions in very low background fractions were measured on a non-air gas matrix using the Picarro CRDS analyzer.

During the last temperature experiments (Table 1e), a temperature step between 20 °C and 180 °C was added (80 °C). This temperature will serve as an upper limit for further temperature experiments using these cylinders. When the cylinders were heated from 20 °C to 80 °C, a reversible difference of 5.1 µmol mol$^{-1}$ was observed only for $H_2O$ measurements of $N_2$ filling for the steel cylinder. For all other species we have not observed any significant production.

The experiments with $N_2$ is a strong indication that the mechanism involves $O_2$. This suggests combustion reactions at high

temperatures under the presence of $O_2$ and with metal surfaces of the cylinders working as catalysts. Production of $CO_2$, CO and $H_2O$ would very well fit to complete and incomplete combustion reactions, however, the production of $CH_4$ cannot be reasonably explained by such reactions. The fact that all enhancements for the species are well-correlated ($r^2 > 0.8$) points toward processes of linked chemical reactions. Moreover, from our observations, it is clear that this process is irreversible with the exception of most $H_2O$ measurements for the steel cylinder. Another possible scenario might be that, under increasing temper-

ature, although as low as 180 °C, diffusion from the interior to the surface of the metal takes place followed by evaporation from the surface which results in enhancements in the gas phase (Smithells et al., 1935). For the steel cylinder, repeating the experiments mostly resulted in less production of amount substance which might be an indication of depletion. However, due to the lack of more detailed information, we cannot conclude which of these scenarios applies as well as the exact underlying mechanism of this process remains unclear. This investigation is beyond the scope of this study. However, these results do not

prevent continuing to use these cylinders, since 180 °C is not a typical temperature for the utilisation of gas cylinders.

## 4   Discussion

This study encompasses a wide range of experiments for the newly built cylinders. It is crucial to note that during these experiments, the background effect of the cylinders varied because of heating or further manipulations (Table 1d). While presenting the results, significant effort was made to highlight these changes and its chronology (Table 1). Such effects were

clearly detected for the aluminum cylinder since it underwent experiments after the high temperature experiments. The results indicate that through heating, and polishing, which was applied in order to eliminate the layer formed during the ultrasonic cleaning, a more inert surface was formed. This new surface showed significantly less enhancement in the amount fraction of $CO_2$.





Since we performed the experiments with a dual QCLAS system after the high temperature runs, it was only possible to compare results from the aluminum cylinder after heating. Combining the results from both analyzers, we were able to validate the applied strategy for the pressure experiments. Setting the cut-off point determined by the outlet valve parameter resulted in less than 0.1 µmol mol$^{-1}$ enhancement in the amount fraction of $CO_2$ for the aluminum cylinder using the CRDS analyzer.

The independent QCLAS measurements on the aluminum cylinder has not shown any effect down to pressures as low as 150 mbar. Considering the 0.12 µmol mol$^{-1}$ standard deviation of the measurements, the data from both systems are compatible with each other. Under sufficient flow rates through the cell, the measurements from the CRDS analyzer are reliable even at pressures lower than 400 mbar.

After the second cleaning procedure (Table 1d) our results compared reasonably well with the low flow experiments reported
in Schibig et al. (2018) and Brewer et al. (2018). The former setup consisted of higher volume cylinders, and they suggested no difference between the coated and uncoated aluminum cylinders. The enrichment in the $CO_2$ amount fraction was reported as 0.090 ± 0.009 µmol mol$^{-1}$(Schibig et al., 2018). On the contrary, Brewer et al. (2018) reported significant differences between the tested treated and untreated aluminum cylinders in their low flow experiments. Their results for the mixtures in untreated aluminum cylinders with high or low water content were 0.1 µmol mol$^{-1}$ or lower. However, it is important to note that, the
previous studies used different cylinder sizes. Schibig et al. (2018) used 29.5 L cylinders, whereas Brewer et al. (2018) used 10 L cylinders corresponding to various surface to volume ratios. Moreover, these studies used different amount fractions of $H_2O$ of >0.05 µmol mol$^{-1}$ and 10 µmol mol$^{-1}$ in Brewer et al. (2018), and >1 µmol mol$^{-1}$ in Schibig et al. (2018). Due to its higher polarity, $H_2O$ molecules are expected to occupy the available sites on the cylinder surface, and result in less adsorption of $CO_2$ molecules, which is supported by Brewer et al. (2018). Despite this varying conditions both studies did not observe
effects higher than 0.1 µmol mol$^{-1}$ at low flow conditions for untreated aluminum cylinders. On the contrary, both studies have shown effects at high flow condition which cannot be explained by adsorption theory alone.

A significant difference between this study and the previous studies (Leuenberger et al., 2015; Brewer et al., 2018; Schibig et al., 2018) is the onset of adsorption effects. All previous studies have shown that for high pressure cylinders, usage below 20 bar is problematic. This was explained through the Langmuir monolayer isotherm (Langmuir, 1918) and its exponential
behavior at low pressures. On the contrary the cylinders tested in this study showed effects only well below atmospheric pressures. A reason of this different evolution of the observations might be explained with the relatively lower fill pressures (up to 30 bar) in this study. At lower pressures, the surface coverage of the available wall spaces decreases, which corresponds to adsorption of lower amount fractions. Another reason of this behavior can be that the aluminum prototype cylinder used in this study has evolved into a superior cylinder in terms of surface interactions compared to commercial cylinders. It is
also worthwhile to note that the surface roughness of the presented aluminum and steel cylinders are 0.8≤Ra≤1.6, which is presumably smoother than the commercial cylinders in use.



## 5   Conclusions and Outlook

We have reported the first results of the newly built cylinders which were designed to serve as chambers to investigate surface effects. The characterization of the cylinders was done under various pressure and temperature conditions, and measurement procedures were established. The rich dataset presented in this study covers the species $CO_2$, CO, $CH_4$, and $H_2O$ using a
commercial and a novel measurement device.

Pressure experiments covered a wide range from 30 bar until a few mbars. During the pressure experiments with CRDS, no effect was observed for $CH_4$ and CO. Prior to heating and cleaning, aluminum and steel have shown comparable adsorptive effects of 0.38 and 0.57 µmol mol$^{-1}$ for $CO_2$ for the highest fill pressure of 30 bar. The behavior of the aluminum cylinder changed after the applied procedures, corresponding to less than 0.1 µmol mol$^{-1}$ difference in the amount fraction between
the begining and the end of the run. The measurements from CRDS at low pressures were validated by an QCLAS system. The results showed that for pressures above 150 mbar, the enhancements in the measured amount fractions did not exceed 0.12 µmol mol$^{-1}$.

The cylinders were tested at low (–10 °C to 80 °C) and high (20 °C to 180 °C) temperature ranges. Under 80 °C, both the aluminum and the steel cylinder showed limited contaminations. After temperatures of 130 °C, irreversible effects became
predominant. These effects were encountered for all measured species, with the exception of $H_2O$ during some runs. Well correlated productions of $CO_2$, $CH_4$ and CO were observed. The most striking difference measured was as high as 8655.4 nmol mol$^{-1}$ for CO.

The presented measurement setup, and established procedures will further be used. Various commonly used materials will be inserted into this cylinders to test for their surface effects. Moreover, the cylinders will be used to investigate effects of other
important atmospheric trace gases such as halocarbons.





# Appendix A

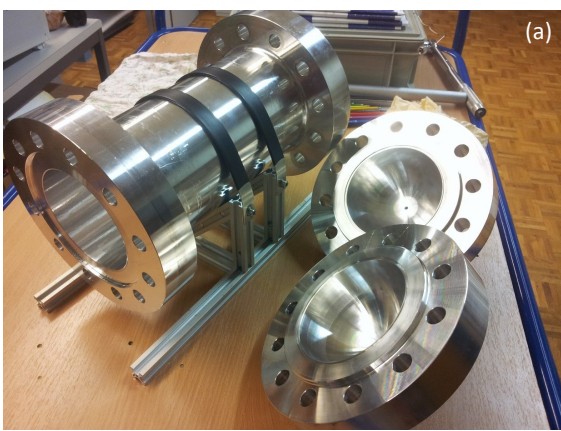 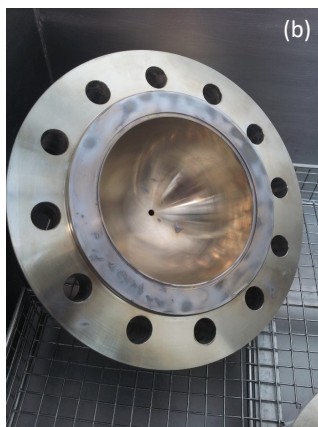

**Figure A1. (a)** Custom made cylinder of three pieces. **(b)** Stains after the ultrasonic bath during the second cleaning procedure (Table 1c).

*Author contributions.* ES, PN, CP, BN and ML designed the CRDS experiments. ES, PN, BB and ML designed the QCLAS experiments. ES carried out the experiments, and PN provided technical support. QCLAS measurements were carried out under the supervision of BB. ES did the data analysis and prepared the manuscript with contributions from all co-authors. ML supervised the project.

5  *Acknowledgements.* This project is supported by a research contract (F-5232.30052) between the Swiss Federal Institute of Metrology (METAS) and the University of Bern. Funding for the development of the QCLAS instrument was provided by the European Research Council (ERC) under the European Union's Horizon 2020 research and innovation programme (grant agreement No 667507 (deepSLice)). The authors would like to thank to the Workshop of University of Bern for the production of the cylinders, and the METAS Gas Analysis Laboratory and METAS workshop for their technical support during this work. The authors would also like to thank to Hubertus Fischer for

10  his valuable comments on the manuscript. The authors are grateful to the members of the Laser Spectroscopy group at Empa for their support during the QCLAS measurements.





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
