# Peer review of "S1 Supplementary material on analysis of CRDS data"

_Atmospheric Measurement Techniques, 2019_

## Referee Comment (RC1) · P. P. Tans (Referee) · 25 Jul 2019

General comments: The authors performed a series of experiments to learn more about wall effects in aluminum and steel high pressure gas cylinders at different pressures and temperatures. The trace gases considered are CO2, CH4, CO and low amounts of water vapor in air. In order to increase wall effects they chose to make special small cylinders with a higher wall to volume ratio. Additional advantages are that one has easy access to the interior surface and it is also easier to control the temperature of the small cylinders in a small oven. However, it is a significant disadvantage that their internal surface may not be the same as in the larger Luxfer cylinders that are

almost universally used to distribute calibration mixtures for high precision greenhouse gas measurements. Luxfer claims that it has a proprietary version of the 6061 alloy, its manufacturing process is very different, and the surface treatment of the author's cylinders is also different from Luxfer's. The smallest high pressure Luxfer cylinder has a volume of only ~700 cc; It is a pity that they did not include it in their experiments. The author's steel cylinder offers a comparison because its wall effects are different from aluminum. Stainless steel is often used for trace gases other than the main greenhouse gases.

Specific comments: page 3 line 32 I wonder why the experiments did not go to 130 bar, at which pressure calibration gas mixtures are often distributed. The highest pressure was only 30 bar, not far above the recommended low pressure use limit of 20 bar. p.4 line 12 This paragraph needs more detail, about the polishing material, what's in the ultrasonic cleaning solution, and then later the "organic agent", and "mild detergent". The stains mentioned have deposited something on the surface, but the elimination of the stains may have deposited something else later. p.7 section 2.3 What is the purpose of going to these low pressures, other than the small size of ice core samples? The section seems to be somewhat out of place with the rest of the experiments. Section 3.1.1 The CRDS analyzer has been used outside of its recommended range, where it cannot regulate its flow and pressure any more. There is a long description, incl. Fig. 3, of how to push a little below the factory-recommended lowest pressure. But is that relevant? Does the adsorption/desorption effect show up between the (absolute) pressures of ~1.4 and ~1.2 bar? Does any calibration gas user insist on going that low? My recommendation is to just stop at 1.4 bar, and shorten this section. It also would make the paper easier to read. Section 3.1.2 I would like to thank the authors for their honest reporting, I wish more people would do that. However, also the one retained filling is a bit worrisome. Why is the response non-linear, both above and below the standard target pressure of 5 mb? Is the absorption line partially saturated? Also, in the correction formula the fitted coefficient "c" (which corresponds to a constant offset between samples and standard) has been omitted. p. 12 line 12 Note that

Schibig found that even at 150 bar pressure only a relatively small fraction of available adsorption sites was occupied. p.19 line 17 There is an important typo here. The ">" symbol should be changed to "<" (less than) in both cases. p. 19 last paragraph needs re-formulation. It now suggests that the authors have lost sight of Schibig's observation that the Langmuir adsorption effect is only ∼0.01 ppm at 75 bar, ∼0.02 ppm at 45 bar, and 0.03 ppm for the 20 bar suggested cutoff. If one's starting pressure is 30 bar, significant effects are not expected above ∼4 bar, and still lower for lower starting pressures. Also the word "problematic" is an overstatement: The high reproducibility of Schibig's results suggest that one could correct for adsorption effects. Finally, the second cleaning may have done some good, but I am not sure that is practical for the large cylinders that are mostly used.

---

## Referee Comment (RC2) · Anonymous Referee #2 · 26 Jul 2019

General comments:

This paper investigates the trace gas stability of air stored in high pressure steel and aluminum cylinders with respect to adsorption/desorption surface processes. Experiments were designed to look at gas phase changes in $CO_2$, $CH_4$, $CO$ and $H_2O$ as functions of gas pressure and temperature.

The matter of trace gas stability in air standards is critical to atmospheric measurement programs. A better understanding of how surface processes affect trace gas concentrations could lead to better selection of cylinder materials and operating procedures.

This paper addresses these matters. It gives detailed descriptions of the experiments

performed, which are new and informative, but the conclusions are somewhat vague. I am left with some unanswered questions. To what extent are the results consistent with the Langmuir adsorption model? The QCLAS measurements at sub-ambient pressures were presumably done to test the adsorption model under extreme pressure conditions. Did the experimental results support the model? Was the observed temperature dependency consistent with the model?

Four different gases were measured but analysis of the results focuses on $CO_2$. This may be because $CO_2$ showed the strongest signals, but the authors should comment on why this is the case, and provide some more discussion of what the results say about the other gases. For example, do the gases differ in their sensitivity to adsorption on account of their molecular properties? Can this explain the different pressure and temperature dependencies observed for the different gases? What conclusions can be drawn for how air standards should be prepared and used?

I think the paper could be suitable for publication in AMT if these questions are addressed.

Specific comments:

It is unclear in some parts of the text if quoted gas pressures are absolute or relative to ambient atmospheric pressure. There is potential for more confusion when referring to cylinder and cell pressures.

Page 8, line 4 – It should be noted here that $CH_4$ decreased while the other three gases increased in concentration. What does this say about the favored explanation of outgassing?

Page 12, line 2 – It is claimed that CO and $CH_4$ dependency on pressure was not significant, but in Figure 6 for $CH_4$ at least, the "Steel before heating" and "Aluminum after heating" plots show elevated $CH_4$ at low pressures. Is this an analytical artefact or a real bias? If real, it requires some comment. The authors should also comment

on why there is a clear effect for CO2 and H2O but not for CO and CH4.

Page 15, Section 3.3.2 – It appears that all four gases are correlated in their response to temperature changes. If so this should be made clear. Is there a reason why the figures and the table consider only correlations between the pairs CO2 – CH4 and CO – H2O?

The readability of the paper is fairly good, but could be improved in places with a little attention from a proficient English speaker. Maybe the editor could help with this. Some specific suggestions are included below.

Technical comments:

Page 1, line 9 – replace "until pressures as low as 150 mbar" with "for pressures down to 150 mbar"

Page 1, line 17 – reword to "measurements of CO2 were made at Mauna Loa, Hawaii in the late 1950s..."

Page 1, line 19 – "with an increasing number"

Page 1, line 21 – Global Atmosphere Watch

Page 2, line 13 – has received attention

Page 2, line 18 – amount fractions

Page 2, line 25 – gas cylinder usage

Page 3, line 3 – reword to "enables placement of test materials..."

Page 3, line 31 – interpret

Page 4, lines 1-2 – Provide some more detail about the flow rate used and the length of time required to obtain reliable measurements. Was there significant change in cylinder pressure?

Page 5, line 7 – I'm not sure that "contamination" is the right word to use here, but that may depend on what caused the stains. Can the authors comment on this?

Page 5, line 10 – replace "until 0.05 mbar" with "to 0.05 mbar"

Page 20, line 10 – beginning

Page 20, line 14 – "Above temperatures of"; again, consider if "contaminations" is the best word to use

---

## Author Comment (AC2) · 11 Sep 2019

The replies to the review was uploaded in the form of a supplement.

Please also note the supplement to this comment:
https://www.atmos-meas-tech-discuss.net/amt-2019-197/amt-2019-197-AC2-supplement.pdf

———————————————————

---

## Author Response (AR1)

**Reply to the review of P.P. Tans**

The authors would like to thank P.P. Tans for his valuable comments. In the following, referee's comments are given in bold and author's responses in plain text. Suggested new text is quoted in italics together with page and line numbers.

**General comments: The authors performed a series of experiments to learn more about wall effects in aluminum and steel high pressure gas cylinders at different pressures and temperatures. The trace gases considered are $CO_2$, $CH_4$, CO and low amounts of water vapor in air. In order to increase wall effects they chose to make special small cylinders with a higher wall to volume ratio. Additional advantages are that one has easy access to the interior surface and it is also easier to control the temperature of the small cylinders in a small oven. However, it is a significant disadvantage that their internal surface may not be the same as in the larger Luxfer cylinders that are almost universally used to distribute calibration mixtures for high precision greenhouse gas measurements. Luxfer claims that it has a proprietary version of the 6061 alloy, its manufacturing process is very different, and the surface treatment of the author's cylinders is also different from Luxfer's. The smallest high pressure Luxfer cylinder has a volume of only~700 cc; It is a pity that they did not include it in their experiments. The author's steel cylinder offers a comparison because its wall effects are different from aluminum. Stainless steel is often used for trace gases other than the main greenhouse gases.**

We appreciate our reviewer's valuable comments and ideas. Unfortunately, we were not aware of the availability of the small size Luxfer cylinders. The small cylinder from Luxfer would indeed be a very valuable addition to these measurements, unfortunately we won't be able to conduct more measurements within the presented study. However, in our study, we put a strong focus on being able to open and close the cylinders because these measurement chambers were constructed primarily for material studies (Satar et al., 2019).

The smaller volume enables to fill and measure the cylinders easily. The surface of the small cylinder $A_{cyl}$= 0.18 $m^2$ which results in a surface to volume ratio of 35.7. Therefore, we estimate the small cylinders to be more susceptible to adsorption by about 40 % than the 29.5 L Luxfer cylinders. It is also crucial to note that the real surface is expected to be significantly larger than the geometric surface area depending on the surface roughness.

We will add the following to the discussion:
*"In order to understand the differences between the constructed cylinders and the Luxfer aluminum cylinder, measurements with a Luxfer cylinder of a similar size (5 L) and pressure ranges (up to 30 bar) would be very useful."*

**Specific comments: page 3 line 32 I wonder why the experiments did not go to 130 bar, at which pressure calibration gas mixtures are often distributed. The highest pressure was only 30 bar, not far above the recommended low pressure use limit of 20 bar.**

The reviewer is right we should have conducted the measurements at higher pressures than 30 bar. However, the current equipment of our system would allow only pressures up to 68.9 bar (limited by the Swagelok valve). The selection of the valve was related to the condition that it does not include

any polymer parts (i.e. full metal valve). It would indeed be useful within further experiments to fill these cylinders higher than 30 bar. Our aim in this study was to establish a measurement and filling procedure and do the first characterization of this newly made cylinders. From these measurements we obtain a slope of 0.01 $\mu$mol mol$^{-1}$ bar$^{-1}$, which would lead to a 1.5 $\mu$mol mol$^{-1}$ enrichment for a cylinder filled to a pressure of 150 bars and decanted to a final absolute cylinder pressure of 400 mbar. This is significantly higher than observed by Schibig et al. (2018) for instance. Therefore, we argue that at 30 bars filling pressure we are close to the maximal adsorption conditions (i.e. close to the measurable $CO_2$ amount fraction), as the Langmuir model says. Moreover, it is worthwhile to conduct experiments at lower pressure ranges (working pressure of analyzers) and increasing fill pressures step by step in order to understand fill pressure dependency.

For clarity we include this information in the manuscript on page 3 after line 27:
*"Although the cylinders were constructed to withhold pressures up to 130 bar, the current setup would enable filling the small cylinders up to 68.9 bar which is limited by the valves (SS-4H from Swagelok). Since this study focused on the first characterization and the establishment of measurement and filling procedures of this newly made cylinders, we present experiments up to 30 bar only."*

**p.4 line 12 This paragraph needs more detail, about the polishing material, what's in the ultrasonic cleaning solution, and then later the "organic agent", and "mild detergent".**

The commercial ultrasonic cleaning solution used in the first ultrasonic-bath is Deconex HT1201 (pH~9.4). The solution is used to remove oil, grease and residues of polishing compounds. However, we believe that either the cleaning agent or the oil residues which were still on the surface resulted in contamination during the temperature experiments. Therefore, we decided to do a second cleaning procedure. In the second ultrasonic bath a relatively neutral detergent (pH~7-8) was used since the alkaline solution (Deconex HT1201) was thought to be too aggressive. The organic agent is a chemical polishing material which was suited for aluminum surfaces.

On page 3, line 7, the following will be added:
*"… a mildly alkaline commercial cleaning agent (Deconex HT1211, pH~9.4)"*

On page 4, line 14, the following will be rephrased:
*"Firstly, the aluminum cylinder was opened and placed in an ultrasonic-bath with a relatively neutral detergent (pH~7-8) and tap water, however the ultrasonic bath cycles at 60 °C ended with further contamination and visible stains (Fig. A1.b). To eliminate this, the two caps were polished with a chemical polishing material which was suited for aluminum surfaces…"*

**The stains mentioned have deposited something on the surface, but the elimination of the stains may have deposited something else later.**

We agree with the reviewer that the elimination of the stains might have deposited something else on the surface of the cylinder. However, our experience was that the aluminum cylinder showed better performance after the elimination of stains without any visible changes in the surface.

**p.7 section 2.3 What is the purpose of going to these low pressures, other than the small size of ice core samples? The section seems to be somewhat out of place with the rest of the experiments. Section 3.1.1 The CRDS analyzer has been used outside of its recommended range, where it cannot regulate its flow and pressure any more. There is a long description, incl. Fig. 3, of how to push a little below the factory-recommended lowest pressure. But is that relevant? Does the adsorption/desorption effect show up between the (absolute)pressures of~1.4 and~1.2 bar? Does any calibration gas user insist on going that low? My recommendation is to just stop at 1.4 bar, and shorten this section. It also would make the paper easier to read.**

We highly appreciate our reviewer's comments on this section. However, in our opinion it is useful to include the measurements from the QCLAS analyzer, since an independent measurement device is a valuable addition for the interpretation of our current results. Moreover, presenting the lower pressure ranges is also useful for other gas applications including development of measurement systems. Although calibration gases are not used at such low limits, the aim of this study is to understand adsorption / desorption processes in its full extent including low pressures where adsorption effects should follow an exponential path.

Regarding the CRDS analyzer, our observations highlight possible systematic errors related to pressure and flow. Reporting such observations are valuable for the understanding of the measurement devices.

**Section 3.1.2 I would like to thank the authors for their honest reporting, I wish more people would do that. However, also the one retained filling is a bit worrisome. Why is the response non-linear, both above and below the standard target pressure of 5 mb? Is the absorption line partially saturated?**

We are confident that after our trials with the aluminum cylinder, we have established a successful procedure and used this setup for further measurements. Since these measurements were conducted after loading material blocks to the aluminum cylinder, they are not presented within this study. The runs with steel loading were reproducible, therefore, we think that presenting the one retained filling for the empty cylinder is non-problematic.

The non-linear response might indeed be related to the relatively high absorbance of the target lines (we observed this issue on two $^{12}CO_2$ lines, one of which was close to saturation which led to a significant offset in the carbon isotope ratio $^{13}C/^{12}C$). Therefore, we have selected the one which was further away from saturation, hence a saturation influence is less probable. A mismatch between the fitting model and the effective profile, crosstalk from the background or a combination of these might explain the response. It should as well be taken into consideration that the set pressure differences were large, corresponding to an order of magnitude change in the lower end (0.5 mbar).

**Also, in the correction formula the fitted coefficient "c" (which corresponds to a constant offset between samples and standard) has been omitted.**

The pressure correction was done relative to the cell pressure. Therefore, in Eqn. 2 (page 11, line 14) the constants (c) cancel each other.

**p. 12 line 12 Note that Schibig found that even at 150 bar pressure only a relatively small fraction of available adsorption sites was occupied.**

We rephrase the sentences in page 12 line 12 for clarity:

*"The aluminum cylinder was in a pressure range (up to 30 bars) where most of its available sites for adsorption were unsaturated. This is in line with the observations of Schibig et al. (2018), which states that even at 150 bar pressure only a relatively small fraction of available adsorption sites was occupied. Changes between 30 and 150 bars seem to be minimal due to the shape of the adsorption isotherm."*

**p.19 line 17 There is an important typo here. The ">" symbol should be changed to "<" (less than) in both cases.**

We thank our reviewer for his attention. The signs are changed accordingly.

**p. 19 last paragraph needs re-formulation. It now suggests that the authors have lost sight of Schibig's observation that the Langmuir adsorption effect is only~0.01 ppm at 75 bar, ~0.02 ppm at 45 bar, and 0.03 ppm for the 20 bar suggested cutoff. If one's starting pressure is 30 bar, significant effects are not expected above ~4 bar, and still lower for lower starting pressures. Also the word "problematic" is an overstatement: The high reproducibility of Schibig's results suggest that one could correct for adsorption effects. Finally, the second cleaning may have done some good, but I am not sure that is practical for the large cylinders that are mostly used.**

We thank our reviewer for his insights. However, the results presented in this study have not followed the shape of the observations of Schibig et al. (2018). This difference is highly likely due to different cylinder properties used in these studies. Please see the replies to anonymous referee #2 for the discussion on K values. It would be worthwhile to investigate the adsorption using the flow-through approach. This would indicate whether the adsorption occurs already at very low pressure or not.

On page 19 line 24, we will change the word *"problematic"* to *"not recommended"*.

On page 20 after line, the following sentence will be added:

*"Additionally, the reverse process of desorption will be investigated by using the flow through approach. Such experiments would be valuable to understand whether adsorption already occurs at very low pressures."*

**Reply to the review of Anonymous Referee #2**

The authors would like to thank anonymous referee for the valuable comments. In the following, referee's comments are given in bold and author's responses in plain text. Suggested new text is quoted in italics together with page and line numbers.

**General comments: This paper investigates the trace gas stability of air stored in high pressure steel and aluminum cylinders with respect to adsorption/desorption surface processes. Experiments were designed to look at gas phase changes in $CO_2$, $CH_4$, CO and $H_2O$ as functions of gas pressure and temperature.**

**The matter of trace gas stability in air standards is critical to atmospheric measurement programs. A better understanding of how surface processes affect trace gas concentrations could lead to better selection of cylinder materials and operating procedures.**

**This paper addresses these matters. It gives detailed descriptions of the experiments performed, which are new and informative, but the conclusions are somewhat vague. I am left with some unanswered questions.**

**To what extent are the results consistent with the Langmuir adsorption model?**

We agree with our reviewer that this point needs further clarification. Our findings did not support the shape of the Langmuir adsorption isotherm as observed in the previous studies (Leuenberger et al. 2015, Brewer et al. 2018, Schibig et al 2018). The onset of the surface effects was not observed until sub-atmospheric pressures for the cylinders tested in this study.

In order to investigate whether the observed amount fraction changes can be explained by the Langmuir adsorption isotherm for monolayer coverage, we used a modified version of the Eqn. 5 from Leuenberger et al. (2015):

$$CO_{2,meas} - CO_{2,initial} + CO_{2,ads} = \Delta CO_2 = CO_{2,ads} \cdot \left( \frac{K \cdot (P - P_0)}{1 + K \cdot P} + (1 + K \cdot P_0) \cdot ln\left( \frac{P_0 \cdot (1 + K \cdot P)}{P \cdot (1 + K \cdot P_0)} \right) \right)$$

Where, $CO_{2,initial} + CO_{2,ads}$ is the mean of the measured amount fractions during the first hour for each experiment. Therefore, for P close to $P_0$, the left side of the equation will be close to zero and it increases with lower pressures. The left term on the right hand side of the above equation is always negative and the right term always positive. Increasing K or $CO_{2,ads}$ values increases the left term. Yet K increase is less pronounced compared to $CO_{2,ads}$ change. K determines the curvature whereas $CO_{2,ads}$ just stretch or compress the values.

In order to find the best possible fit, we have used R's inbuilt "optim" function with the setting Limited-memory Broyden-Fletcher-Goldfarb-Shanno (BFGS) algorithm. Upper and lower bounds were set for each unknown ($CO_{2\ ads}$ and K) and the algorithm was run to minimize the sum of squared differences between the measured amount fractions and the modelled amount fractions. For $CO_{2,ads}$ lower and upper boundaries were set as 0.001 μmol mol$^{-1}$ and 15 μmol mol$^{-1}$. We have set the first guess values for the algorithm to the lower boundaries. In Figure 1, we show the theoretical isotherms together with our experimental data. The purple points show measurement data from the 30 bar experiments of the aluminum cylinder, and the black lines show the Langmuir monolayer fit to

the measurements with K values of 0.001 bar$^{-1}$ and 1 bar$^{-1}$ denoted by the solid and dashed lines, respectively. Fig 1.b shows a zoom-in to the region where the pressure in the cylinder is less than 3 bar. In order to find a better fit to the experimental data, we have further increased the upper limit of the K value up to 500 bar$^{-1}$ (Table 1). At higher K values, the modelled curve fits better to the onset of the increasing amount fractions. The tendency of a higher K value in this study, contradicts to Schibig et al. (2018), where they have set the K value at 0.001 bar$^{-1}$. The difference between the estimated equilibrium constants in this study and Schibig et al. (2018) may be explained through the different surface properties (e.g. roughness or treatment of surface). Moreover, even by setting larger limits for all parameters, we did not find any K value which was able to fit the highest enrichments measured towards the end of the experiment. This might partly be related to the algorithm we have used and the limited number of data at the end of the measurements. However, the discrepancy for the highest enrichments can also be explained by another effect than desorption at low pressures. The reasonable range of K remains unclear. A more detailed analysis on model fitting is not within the scope of this experimentally focused study.

[Figure]

**Figure 1:** (a) Measured and modelled amount fractions of $CO_2$ for the aluminum cylinder filled to 30 bar. Purple points show measured data, black solid lines show the fit with K=0.001 bar$^{-1}$, black dashed lines show the fit K=1 bar$^{-1}$, dark red lines show the fit with K=10 bar$^{-1}$, and orange dotted lines show the fit K=100 bar$^{-1}$, and black long dashed lines show the fit K=379 bar$^{-1}$ (b) Zoom-in to the region where the cylinder pressure is less than 3 bar.

**Table 1:** Model parameters for Langmuir adsorption isotherm for CRDS data

| K (bar$^{-1}$) [1] | $CO_{2,\,ads}$ (µmol mol$^{-1}$) [2] |
| --- | --- |
| 0.001 | 0.029 |
| 1 | 0.015 |
| 10 | 0.038 |
| 100 | 0.301 |
| **379 [3]** | **1.116** |

[1] Upper boundary for K is increased from 0.001 bar$^{-1}$ to 500 bar$^{-1}$ stepwise for each solution

[2] Lower and upper boundaries for $CO_{2,\,ads}$ 0.001 µmol mol$^{-1}$ and 15 µmol mol$^{-1}$

[3] The best fit was not limited by the boundary conditions.

The following statement will be added at page 19 line 25:

*"In contrast to the previous studies, the cylinders tested in this study showed enrichments only well below atmospheric pressures for the steel cylinder and the aluminum cylinder before heating. At sub atmospheric pressures, the enrichments followed a steep increase. This increase can only partly be fitted to the Langmuir adsorption isotherm if the equilibrium constant (K, the ratio between adsorption and desorption rates) are set to values higher than 1 (Supplementary material). Higher K values would correspond to higher surface coverage factors even at lower fill pressures. In comparison Schibig et al. (2018) have fixed the K value at 0.001 bar$^{-1}$, corresponding to lower surface coverage even at pressures of 150 bar. The reasonable range of the equilibrium constant remains unclear. The differences in the cylinder interior characteristics such as surface roughness or treatment is highly likely the explanation of the discrepancy in the K values. A further investigation on the K value and modelling approaches is not within the scope of this experimental focused study."*

Figure 1 will be added to the supplementary material together with the information on the method used for the fit.

**The QCLAS measurements at sub-ambient pressures were presumably done to test the adsorption model under extreme pressure conditions. Did the experimental results support the model?**

Indeed, these measurements were conducted to test the adsorption model under extreme conditions. Moreover, these measurements can also be useful for measurement systems operated at low pressure conditions. The experimental results supported the Langmuir model, however, the enrichments occurred in the region where the pressure correction function required extrapolation (page 11 line 19). Therefore, these data should be interpreted carefully. Our aim when conducting the QCLAS experiments was to find the lower limit under which CRDS measurements would be reliable. Nevertheless, we have conducted a similar analysis as presented above in order to determine model fit parameters. We add the Langmuir fit (Figure 2) to supplementary material.

[Figure]

**Figure 2:** Measured and modelled amount fractions of $CO_2$ for the aluminum cylinder from the QCLAS setup. Red points show measured data, black dashed lines show the modelled fit with K=0.001 bar$^{-1}$, black lines show the modelled fit with K=0.01 bar$^{-1}$, blue lines show the modelled fit with K=0.152 bar$^{-1}$ and black dotted lines show the modelled fit with K=1 bar$^{-1}$

**Table 2:** Model parameters for Langmuir adsorption isotherm for QCLAS data

| K (bar$^{-1}$) [1] | $CO_{2,\,ads}$ (μmol mol$^{-1}$) [2] |
|---|---|
| 0.001 | 0.185 |
| 0.01 | 0.149 |
| **0.152 [3]** | **0.454** |
| 1 | 2.387 |

[1] Upper boundary for K is increased from 0.001 bar$^{-1}$ to 1 bar$^{-1}$ stepwise for each solution

[2] Lower and upper boundaries for $CO_{2,\,ads}$ 0.001 μmol mol$^{-1}$ and 15 μmol mol$^{-1}$

[3] The best fit was not limited by the boundary conditions.

**Was the observed temperature dependency consistent with the model?**

The observed temperature dependency was not consistent with the Langmuir adsorption isotherm at least above 80 °C. The temperature dependencies observed in the presented study are irreversible and not related to a physical adsorption. Within the scope of other studies (Leuenberger et al., 2015 and unpublished data), reversible temperature responses are measured until 80 °C, however these differences were an order of magnitude smaller than the presented enrichments in this study at 180 °C. Therefore, we are confident that the observed differences, above 80 °C are not related to a reversible adsorption process. The discussion on page 18 starting from line 14 explains other relevant hypotheses on the observed enrichments.

**Four different gases were measured but analysis of the results focuses on CO$_2$. This may be because CO$_2$ showed the strongest signals, but the authors should comment on why this is the case, and provide some more discussion of what the results say about the other gases. For example, do the gases differ in their sensitivity to adsorption on account of their molecular properties? Can this explain the different pressure and temperature dependencies observed for the different gases?**

We agree with our reviewer that more information on other species is necessary. The reason that our study concentrate on $CO_2$ is indeed related to the strong amount fraction response of $CO_2$.

The following paragraph will be added at page 19 after line 31:

*"This study also showed that the measured gases CO, CO$_2$, CH$_4$ and H$_2$O had different sensitivities with respect to surface processes. We have observed surface effects for CO$_2$ and H$_2$O. Observed effects of H$_2$O during the pressure experiments were an order of magnitude larger than CO$_2$ (not shown here). One of the explanations that CO$_2$ and H$_2$O are more prone to surface effects might be due to their high boiling points. CO$_2$ sublimates at -78.5 °C, and the boiling point of H$_2$O is 100 °C. Whereas for CH$_4$ and CO, boiling points are -161 ° C and -191.5 °C, respectively. Since CO is a reactive compound, it might be argued that it would be more prone to surface effects. However, our results have shown that CO in atmospheric air was not affected by surface interactions at short time scales (in the order of days). This is highly likely related to the competitive adsorption between species. The ratio between the amount fraction of CO and CO$_2$ would be 1 to several hundreds. In order to understand competitive adsorption to its full extent, experiments focusing on a range of amount fractions would be useful. Moreover, when discussing adsorption properties, polarity is also an*

*important criterion. Therefore, the non-polar structure of $CH_4$ makes it less prone to adsorption, whereas the polar geometry of $H_2O$ enables it to be more adsorptive."*

**What conclusions can be drawn for how air standards should be prepared and used?**

The air standards should not be stored at high temperatures. However, high temperatures might be useful for pre-treatments of cylinders. Aluminum cylinders are well suited to store greenhouse gases such as CO, $CO_2$ and $CH_4$, whereas usage of stainless steel cylinders are more suited for standards of halogenated compounds.

**I think the paper could be suitable for publication in AMT if these questions are addressed.**

**Specific comments:**

**It is unclear in some parts of the text if quoted gas pressures are absolute or relative to ambient atmospheric pressure. There is potential for more confusion when referring to cylinder and cell pressures.**

We agree with our reviewer this point need clarification. Cell pressures are consistently reported as absolute pressures, whereas cylinder fill pressures for CRDS measurements are consistently reported relative to the ambient pressure. We make the following additions to the manuscript in order to prevent confusion:

On page 1, line 9:
*"This extensive dataset revealed that for absolute pressures down to 150 mbar the enhancement in the amount fraction of $CO_2$ relative to its initial value (at 1200 mbar absolute) ..."*

In Table 1 on page 6,
*"[bar relative to atm]"*

On page 7, line 16:
*"Therefore, we filled the aluminum cylinder to 1200 mbar (absolute)"*

On page 10, caption of Figure 5:
*"Reported pressure data show absolute pressure values."*

On Page 11, line 17:
*"At the point where the cell pressure started to fall below the target pressure (150 mbar absolute-Fig. 5c) ..."*

On page 12, caption of Figure 6:
*"x-axes show the absolute pressure values in the sample cylinder."*

On page 19, line 5:
*"The independent QCLAS measurements on the aluminum cylinder has not shown any effect down to absolute pressures as low as 150 mbar."*

On page 20, line 11:
"The results showed that for absolute pressures above 150 mbar… "

**Page 8, line 4 – It should be noted here that CH$_4$ decreased while the other three gases increased in concentration. What does this say about the favored explanation of outgassing?**

Our reviewer points out an interesting point, however, the underlying mechanism of the instrument related effects are unclear. The decrease in CH$_4$ might partly be related to a dilution caused by the increase of other compounds in the cavity.

**Page 12, line 2 – It is claimed that CO and CH$_4$ dependency on pressure was not significant, but in Figure 6 for CH$_4$ at least, the "Steel before heating" and "Aluminum after heating" plots show elevated CH$_4$ at low pressures. Is this an analytical artefact or a real bias? If real, it requires some comment. The authors should also comment on why there is a clear effect for CO$_2$ and H$_2$O but not for CO and CH$_4$.**

We thank the reviewer for pointing this out. We relate these changes to an analytical artefact likely related to a drift in CH$_4$ measurements and the cavity pressure instabilities which occurred towards the end of the experiments. Since the onset of this increase is not the same for the conducted experiments and the observed differences are not consistent through the replicates, we do not relate these effects to adsorption / desorption processes. Fig. 7c clearly shows the differences among the replicates of steel cylinder, and the aluminum after heating experiments.

Please see the text above for explanation of the adsorptive properties of all measured species.

**Page 15, Section 3.3.2 – It appears that all four gases are correlated in their response to temperature changes. If so this should be made clear. Is there a reason why the figures and the table consider only correlations between the pairs CO$_2$ – CH$_4$ and CO– H$_2$O?**

We agree with the reviewer that this point needs clarification. Indeed, all pairs are correlated, for easier visibility only two pairs at a time was shown. The species were paired by highest coefficient of determinations.

The reason behind such correlations might be chemical reactions following fixed ratios of production, however it is highly questionable if it is feasible to produce methane under 180 °C and slightly over 10 bar. This is already explained on page 18 lines 14-18.

**The readability of the paper is fairly good, but could be improved in places with a little attention from a proficient English speaker. Maybe the editor could help with this. Some specific suggestions are included below.**
We thank our reviewer for his attention, the technical corrections noted below are changed at the respective places.

**Technical comments:**
**Page 1, line 9 – replace "until pressures as low as 150 mbar" with "for pressures down to 150 mbar" –** text modified accordingly

**Page 1, line 17 – reword to "measurements of CO2 were made at Mauna Loa, Hawaii in the late 1950s..." –** text modified accordingly

**Page 1, line 19 – "with an increasing number" -** corrected

**Page 1, line 21 – Global Atmosphere Watch -** corrected

**Page 2, line 13 – has received attention -** corrected

**Page 2, line 18 – amount fractions -** corrected

**Page 2, line 25 – gas cylinder usage -** corrected

**Page 3, line 3 – reword to "enables placement of test materials..." -** corrected

**Page 3, line 31 – interpret -** corrected

**Page 4, lines 1-2 – Provide some more detail about the flow rate used and the length of time required to obtain reliable measurements. Was there significant change in cylinder pressure?**

The flow rate during the pressure experiments is shown in Fig. 4a. The flow rate into the cell of the CRDS analyzer is between 15 mL min$^{-1}$ and to 220 mL min$^{-1}$, regulated by the outlet valve as explained in Sect. 3.1.1. Regarding the time required, the measurement setup has 1/4" tubing which is 30 cm long. Prior to the experiment, the tubing and the pressure regulator were flushed 3 times. For the analysis the first 10 minutes of data was not taken into consideration. 10 minutes of measurements with 220 mL min$^{-1}$ would correspond to a 0.4 bar decrease in the pressure of the small cylinder. Since the observed effects in the cylinders does not start until pressures less than atmospheric pressures, the change is not significant for the presented experiments.

For clarity we include this information also in Page 4 line 3:

[revised manuscript text omitted]